# Mechanistic Links Between Gut Dysbiosis, Insulin Resistance, and Autism Spectrum Disorder

**DOI:** 10.3390/ijms26136537

**Published:** 2025-07-07

**Authors:** Patricia Guevara-Ramírez, Rafael Tamayo-Trujillo, Viviana A. Ruiz-Pozo, Santiago Cadena-Ullauri, Elius Paz-Cruz, Ana Karina Zambrano

**Affiliations:** Centro de Investigación Genética y Genómica, Facultad de Ciencias de la Salud Eugenio Espejo, Universidad UTE, Quito 170129, Ecuador

**Keywords:** healthcare, autism spectrum disorder, microbiota, insulin resistance, molecular pathways

## Abstract

Autism spectrum disorder (ASD) is a neurodevelopmental condition frequently associated with gastrointestinal symptoms, gut dysbiosis, and metabolic dysfunctions such as insulin resistance (IR). Recent evidence suggests that the gut microbiota may influence both metabolic and neurological processes through the gut–brain–metabolic axis. This review explores the molecular mechanisms linking dysbiosis, IR, and ASD, focusing on pathways such as TLR/NF-κB activation, PI3K/Akt/mTOR disruption, and the action of microbial metabolites, like short-chain fatty acids (SCFAs), lipopolysaccharide (LPS), and γ-aminobutyric acid (GABA). We discuss how dysbiosis may contribute to increased intestinal permeability, systemic inflammation, and neuroimmune activation, ultimately affecting brain development and behavior. Common microbial alterations in ASD and IR—including increased Clostridium, Desulfovibrio, and Alistipes, and reduced Bifidobacterium and butyrate-producing genera—suggest a shared pathophysiology. We also highlight potential therapeutic strategies, such as microbiota modulation, insulin-like growth factor 1 (IGF-1) treatment, and dietary interventions. Understanding these interconnected mechanisms may support the development of microbiota-targeted approaches for individuals with ASD metabolic comorbidities.

## 1. Introduction

Autism spectrum disorder is a complex neurodevelopmental condition that typically manifests in early childhood and persists throughout life [1]. It is characterized by a wide range of symptoms that vary between individuals and are generally grouped into behavioral and medical categories [1]. People with ASD often exhibit deficits in social communication and interaction, accompanied by restricted and repetitive patterns of behavior, interests, or activities [2]. This broad heterogeneity poses significant challenges for diagnosis, management, and the development of effective interventions [1].

Epidemiological data reflects a rising prevalence of ASD worldwide. According to the most recent estimates from the U.S. Centers for Disease Control and Prevention (CDC), approximately 1 in 31 children aged 8 years were identified with ASD in the United States in 2022 [3]. A systematic review covering data from 2008 to 2024 reported a global prevalence of 0.77%, with regional variations: 0.28% in Asia, 1.1% in America, 0.71% in Europe, and 1.51% in Africa and Australia [4]. Although discrepancies exist among prevalence studies, such as the one published in 2022 [5], the observed increase is likely attributable to improved awareness and diagnostic practices [4].

Gastrointestinal problems have been reported in approximately one-third of the children with ASD, with common symptoms including constipation, excessive abdominal gas, and diarrhea [1]. These symptoms not only contribute to increased morbidity but are also associated with greater severity of core ASD features. The high prevalence of gastrointestinal issues has drawn significant attention to the gut–brain axis [6]. This axis is a bidirectional communication network linking the gut microbiota, the central nervous system (CNS), and metabolic processes [7].

Growing evidence indicates a strong correlation between the gut microbiota and brain function, suggesting that gut dysbiosis may play a critical role in the development and regulation of the CNS, as well as in the pathogenesis of neurodevelopmental disorders [8]. Several pathways mediate this communication, including the vagus nerve and a range of endocrine, immune, and biochemical signaling mechanisms [8]. Dysbiosis has been linked to neurological disorders through mechanisms such as activation of the hypothalamic–pituitary–adrenal (HPA) axis, imbalances in neurotransmitter production, systemic inflammation, and increased permeability of the intestinal and blood–brain barriers [8].

Furthermore, recent studies have proposed an expanded gut–brain–metabolic axis that integrates metabolic processes, such as glucose regulation, into the microbiota–neurodevelopment framework [7]. Notably, insulin resistance, a metabolic state in which tissues fail to respond adequately to insulin, has emerged as a relevant factor in neurodevelopmental disorders [9]. However, the molecular mechanisms linking insulin resistance, the gut microbiota, and ASD remain poorly understood.

This article aims to explore, from a molecular perspective, the emerging evidence linking insulin resistance and alterations in the gut microbiota within the context of ASD. By integrating current scientific evidence, this article seeks to provide an integrated view of the potential molecular pathways that may underlie this triad. This perspective may offer new insights for targeted interventions and promising directions for future research.

## 2. The Effects of Dysbiosis in ASD

In recent years, the study of the gut microbiome has emerged as a central focus in understanding the biological factors underlying ASD. Several studies have documented specific alterations in the gut microbiome composition of individuals with ASD, generating significant interest in its potential role in the manifestation of both gastrointestinal and neurobehavioral symptoms [10,11]. These findings suggest that the gut microbiota plays a fundamental role in the function and regulation of the CNS through the gut–brain axis. In this context, dysbiosis not only exacerbates gastrointestinal disorders but may also influence neuroinflammation, neurotransmission, and brain metabolism, positioning itself as a significant modulating factor in the pathophysiology of ASD [12].

Dysbiosis can influence neurodevelopment through multiple interrelated pathways, particularly via the synthesis and signaling of neuroactive compounds. Gut microorganisms can produce neurotransmitters such as serotonin (5-HT), γ-aminobutyric acid, and dopamine, as well as key metabolites like short-chain fatty acids and aromatic derivatives, which can directly or indirectly modulate brain circuits [13,14]. Approximately 90% of serotonin is synthesized in the gut from tryptophan by enterochromaffin cells, a process regulated by the gut microbiota through modulation of tryptophan hydroxylase 1 (TPH1) [15]. In ASD, peripheral hyperserotonemia accompanied by central serotonin deficiency has been reported, potentially reflecting impaired serotonergic signaling associated with social and behavioral deficits [16].

Similarly, the gut microbiota contributes to the production of GABA, a key inhibitory neurotransmitter. Bacteria such as Bacteroides, Bifidobacterium, and Lactobacillus produce GABA via glutamate fermentation. Disruptions in their abundance or function may reduce GABA bioavailability in the CNS [17,18]. Altered GABAergic signaling has been implicated in the excitatory/inhibitory imbalance observed in ASD, which may underlie neuronal hyperexcitability, anxiety, and repetitive behaviors [19,20]. Dopaminergic pathways may also be indirectly modulated by microbial activity through effects on tyrosine metabolism, inflammation, and neurotransmitter transporters, such as the dopamine transporter (DAT), with implications for motivation and executive functioning in ASD [21].

Moreover, microbial fermentation of dietary fibers produces SCFAs, such as butyrate and propionate, which exert neuroactive effects by modulating synaptic plasticity, microglial activation, and gene expression. Reduced butyrate levels, commonly observed in the fecal microbiota of children with ASD, may compromise intestinal barrier integrity, alter CNS immune responses, and affect serotonergic and dopaminergic pathways. In mouse models, gut dysbiosis was associated with reduced social behavior, increased fecal butyrate levels, and neutrophil infiltration, suggesting a role for altered microbiota in autism-related behaviors [22].

In contrast, in animal models, excessive propionate levels have been shown to induce ASD-like behaviors, mitochondrial dysfunction, and oxidative stress. Other microbial metabolites, such as 4-ethylphenyl sulfate (4EPS), derived from *Clostridium* species, have demonstrated neurotoxic potential by disrupting myelination and altering oligodendrocyte maturation, which can contribute to emotional dysregulation and social impairments [15,23].

These findings underscore that microbial dysbiosis in ASD is not merely a gastrointestinal manifestation but may act as a critical modulator of neurochemical signaling, neuroimmune responses, and brain development. Understanding the precise microbial and metabolic pathways involved may offer promising directions for novel diagnostic and therapeutic strategies.

## 3. The Role of Insulin Resistance in Neurodevelopment

Insulin and insulin-like growth factor 1 signaling play crucial roles in maintaining brain homeostasis and regulating neurodevelopmental processes [9]. Insulin is implicated in two major signaling cascades, AKT/mTOR and RAS/ERK, which regulate cellular growth, metabolism, and survival. Disruptions in these pathways have been strongly associated with various neurodevelopmental disorders, particularly ASD [24,25]. IGF-1, a neurotrophic factor critical for proper CNS development, is involved in neuronal growth, synaptogenesis, survival, and migration [26]. Through its widely expressed receptor, IGF1R, it acts via endocrine, paracrine, and autocrine mechanisms [26].

Multiple factors can contribute to insulin signaling dysfunction in neurodevelopmental disorders such as ASD, including inflammation, oxidative stress, and mitochondrial dysfunction, all of which can lead to IR [27,28]. Genetic predisposition and environmental factors during brain development may exacerbate these conditions [28].

An additional mechanism of interest involves the receptor for advanced glycation end products (RAGE), which activates the PI3K/Akt/mTOR signaling pathway and promotes inflammatory neurodegeneration [29,30]. Disruption of this pathway has been linked to defects in neuronal translation, resulting in decreased levels of proteins involved in synapse formation or neuronal structure [31]. Similar disruptions are observed in IR, which promotes defects in neuronal autophagy and apoptosis, contributing to neurodegenerative disorders such as Parkinson’s disease [9]. Furthermore, IR has been associated with an exaggerated immune response due to the loss of insulin’s anti-inflammatory effects, leading to increased production of pro-inflammatory transcription factors (NF-κB) and pro-inflammatory cytokines (IL-1β, IL-6, TNF-α), as well as Toll-like receptor (TLR) overexpression [32].

Emerging research has identified a notable association between IR and ASD. One study found that 16.85% of children and adolescents with ASD, treated with risperidone, exhibited IR, independent of pharmacogenetic gene polymorphisms or drug plasma level [33]. Another investigation comparing individuals with ASD to neurotypical controls revealed that Homeostasis Model Assessment of Insulin Resistance (HOMA-IR) scores were 0.31 units higher in the ASD group, even after adjusting for variables such as sex, age, BMI z-score category, and lipids [34]. Moreover, a systematic review explored insulin resistance in children with autism, revealing no conclusive link with prenatal IR, but highlighting postnatal risk factors such as poor diet, inactivity, and antipsychotic use. Elevated HOMA-IR levels in adolescents with ASD suggest altered glucose metabolism, underscoring the need for further research on metabolic–neurodevelopmental interactions [35].

## 4. Gut Dysbiosis Induces Low-Grade Inflammation and Insulin Resistance

Gut dysbiosis has been implicated in low-grade chronic inflammation through the recognition of pathogen-associated molecular patterns (PAMPs), such as lipopolysaccharide or flagellin, by pattern recognition receptors (PRRs), primarily those of the TLR family. This recognition triggers inflammatory responses in the host [36]. Dysbiosis can arise from various environmental and lifestyle factors, including an unhealthy diet [37], antibiotic use [38], and exposure to toxic compounds, microplastics, and pollution, among others [39,40,41].

The gut microbiota plays a crucial role in host immunity modulation, and its dysbiosis is associated with abnormal production of inflammatory biomarkers, such as C-reactive protein (CRP) or interleukin-6 (IL-6) [42]. Inter-individual differences in cytokine profiles often reflect specific microbial compositions that influence host immune responses [43]. For instance, microbial tryptophan metabolism produces the metabolite tryptophol, which has demonstrated inhibitory effects on TNF-α responses [43].

A state of low-grade chronic inflammation is also frequently observed in individuals with type 2 diabetes or IR [44]. Microbial alterations in overweight individuals include a reduction in sulfate-reducing bacteria and Bacteroides, alongside increased production of branched-chain fatty acids, phenolics, valeric acid, and hydroxy acids, all of which contribute to systemic inflammation [45]. Dysbiosis may also impair gut barrier function, facilitating metabolic endotoxemia and exacerbating complications such as retinopathy and nephropathy in obese individuals [46]. Thus, gut dysbiosis, through the overproduction of intestinal pro-inflammatory cytokines, may promote the migration of these inflammatory mediators and bacterial antigens to various host organs, such as the pancreas, inducing local inflammatory processes that could contribute to the development of IR.

### 4.1. Immune Dysregulation and Cytokine Imbalance in ASD

Immune dysregulation and elevated pro-inflammatory cytokine levels have been implicated in the early onset of ASD. For instance, increased interleukin-8 (IL-8) and decreased IL-10 levels have been detected in children with ASD compared to controls [47]. The activation of inflammasomes has also been described in individuals with ASD, promoting the overexpression of IL-1β and IL-18, and a reduction in the anti-inflammatory IL-33 cytokine [48]. Additional evidence has identified increased levels of inflammatory cytokines, including TNF-α, IL-4, and IL-21, in the cerebrospinal fluid of individuals with ASD, further highlighting the role of immune response in ASD pathogenesis and the potential utility of cytokine profiles for differential diagnosis from other neurological disorders [49]. In ASD, the production of these altered cytokines has been linked to gut dysbiosis and gut permeability [48,50]. Another study linked gut dysbiosis to ASD through a microbiota-driven “TNFα–sphingolipid–st–steroid hormone” axis. Children with ASD showed elevated TNF-α, enrichment of *Bifidobacterium bifidum* and *Segatella copri*, and upregulation of sphingolipid metabolism. Metabolomics revealed reduced steroid hormones, including estriol and deoxycorticosterone. TNF-α correlated positively with microbial toxin pathways and negatively with steroid biosynthesis. These findings highlight a potential mechanism by which microbial and immune disruptions contribute to neurodevelopmental alterations in ASD [51].

Although definitive evidence is still lacking, it could be hypothesized that gut dysbiosis may promote the translocation of bacterial components, such as LPSs, from the intestine to the bloodstream, ultimately reaching neuronal tissues. Once there, LPSs may trigger several pro-inflammatory signaling cascades, mainly via TLR pathways, leading to IL overproduction, which could contribute to ASD pathogenesis [48,52] (Figure 1).

### 4.2. TLR-Mediated Inflammatory Signaling Triggered by LPS: A Mechanistic Link Between Intestinal Permeability and ASD Pathogenesis

TLR signaling may potentially be involved with LPS recognition in ASD associated with increased intestinal permeability [48]. This pathway may begin with the recognition of LPSs by TLR2, TLR4, and TLR9 expressed in neuronal cells. This recognition leads to receptor dimerization and the recruitment of adaptor proteins, such as Toll/interleukin 1 receptor domain-containing adapter interferon-β (TRIF) and myeloid differentiation primary-response protein 88 (MyD88). Subsequently, MyD88 binds to IL-1R-associated kinases, which activate tumor necrosis factor receptor-associated factor 6 (TRAF6). Activated TRAF6 then triggers the mitogen-activated protein kinase (MAPK) pathway and nuclear factor kappa B (NF-κB), initiating inflammatory responses. Moreover, TLR3 and 4 promote the translocation of NF-κB to the nucleus and induce the synthesis of interferon-beta (IFN-β), chemokines such as CCL5 and CXCL10, and the expression of IL-1β and IL-18 [29,53].

Thus, the proposed role of increased intestinal permeability in ASD development could be supported by the possibility that LPSs could trigger this inflammatory pathway. Although further research is required, exploring this pathway may reveal new therapeutic and diagnostic opportunities for individuals with ASD.

Consequently, gut dysbiosis that increases intestinal permeability may contribute to the development of ASD by enabling the migration and dissemination of bacterial antigens, such as LPSs, into neuronal tissue, potentially inducing local inflammatory responses that promote neurological dysfunction associated with ASD. Moreover, under conditions of enhanced internal permeability, the translocation of LPSs into pancreatic islets may decrease insulin production, thereby diminishing insulin’s anti-inflammatory effects in the brain, which could further contribute to ASD symptom onset. These hypotheses could be assessed in pregnant animal models exhibiting insulin signaling dysfunction to evaluate whether maternal gut dysbiosis promotes the migration of maternal bacterial antigens into the fetal neuronal tissue, potentially triggering neurodevelopmental disorders such as ASD.

### 4.3. Gut Microbiota-Derived Metabolites and Tissue-Specific Impacts on Insulin Resistance

In parallel, the gut microbiota has gained recognition as a metabolically active “organ” capable of producing a wide variety of bioactive metabolites. These gut bacteria-derived compounds interact with metabolically active organs, playing a central role in regulating glucose and lipid homeostasis, inflammation, and insulin sensitivity [54,55]. Under the conditions of IR, gut dysbiosis leads to altered microbial metabolism, characterized by a decrease in beneficial metabolites, such as SCFAs, and an increase in detrimental compounds, such as trimethylamine N-oxide (TMAO), hydrogen sulfide, and phenylacetic acid [56].

Insulin regulates glucose homeostasis across skeletal muscle, the liver, and adipose tissue [56,57]. In skeletal muscle, insulin promotes glucose uptake. However, under IR conditions, this uptake is compromised by disruptions in the insulin signaling pathway, particularly in the translocation of the GLUT4 transporter. Consequently, glucose not utilized by muscle is redirected to the liver, favoring de novo lipogenesis and ectopic lipid accumulation, thereby exacerbating IR [57,58]. In this context, a study revealed that extracellular vesicles derived from bacteria such as *Pseudomonas panacis* can block insulin signaling in skeletal muscle and adipose tissue, reducing glucose uptake and worsening metabolic dysfunction [59].

The liver plays a central role in glucose metabolism, including gluconeogenesis, glycogenolysis, glycogen synthesis, and de novo lipogenesis [60]. Under healthy conditions, insulin suppresses hepatic glucose production. However, in IR, this suppression fails, leading to a phenomenon called selective hepatic insulin resistance [61]. For instance, hydrogen sulfide, from sulfate-reducing bacteria, enhances gluconeogenesis and inhibits glycogen synthesis [62,63], while phenylacetic acid from *Bacteroides* spp. has been associated with non-alcoholic fatty liver disease (NAFLD), promoting hepatic triglyceride accumulation by inhibiting AKT phosphorylation, a key event in insulin signaling [56,64]. TMAO has also been shown to exert direct effects on hepatocytes and has been linked to insulin resistance and alterations in glucose metabolism [65,66]. In animal models, TMAO activates cellular pathways, such as the endoplasmic reticulum kinase PKR, and induces the expression of gluconeogenic genes (G6pc and Pck1) via the transcription factor FOXO1, contributing to the development of hyperglycemia [67].

Adipose tissue, recognized as both an endocrine organ and an energy reservoir, is also highly responsive to insulin action. Under normal conditions, insulin stimulates glucose and fatty acid uptake and inhibits lipolysis. In IR, these processes are impaired: glucose uptake is reduced, and the release of free fatty acids and glycerol is increased, contributing to lipid accumulation in the liver and muscle and favoring hepatic gluconeogenesis [68,69]. This metabolic dysfunction is further exacerbated by chronic inflammation, driven by macrophage infiltration and the release of pro-inflammatory cytokines [70,71]. Microbial metabolites such as TMAO, SCFAs, and indole derivatives influence adipokine signaling, including leptin and adiponectin, thereby altering metabolic pathways, promoting oxidative stress, and contributing to the progression of IR [70,72].

Among microbial-derived metabolites, SCFAs—particularly acetate, propionate, and butyrate—exert protective effects on host metabolism. These metabolites, produced through dietary fiber fermentation, activate the AMPK pathway, stimulate fatty acid oxidation, and reduce lipid accumulation in the liver. In skeletal muscle, they promote the expression of genes involved in lipid oxidation and glucose uptake [73,74]. In adipose tissue, propionate and butyrate reduce inflammation, and propionate also increases GLUT4 expression, thereby enhancing insulin-induced glucose uptake. Overall, these findings establish SCFAs as essential mediators in maintaining insulin sensitivity and highlight their potential as therapeutic agents for addressing dysbiosis and metabolic diseases [56,64,74].

### 4.4. Differential Abundance of the Gut Microbiota in ASD and Insulin Resistance

In individuals with IR, a distinctive gut microbial profile has been identified compared to metabolically healthy subjects. This profile is characterized by a lower abundance of symbiotic bacteria, such as *Akkermansia muciniphila* [75], *Bifidobacterium* spp. [76], and *Clostridium coccoides* [76], whose presence is associated with increased gut barrier integrity, reduced inflammation, and improved insulin sensitivity. In contrast, species such as *Prevotella copri* and *Bacteroides vulgatus* have been identified as key drivers of the microbial biosynthesis of branched-chain amino acids (BCAAs) [77]. The accumulation of BCAAs in the circulation can interfere with insulin signaling through the mTOR and IRS1 pathways. In addition, this dysbiosis favors the expansion of LPS-producing Gram-negative bacteria, contributing to chronic metabolic endotoxaemia that activates the TLR4/NF-κB pathway. This, in turn, leads to systemic inflammation and dysfunction of insulin-dependent signaling in peripheral tissues [77,78,79].

Complementing this evidence, a recent multi-omics analysis identified functional microbial signatures linked to IR. In insulin-resistant individuals, increased levels of genera such as Blautia and Dorea (family Lachnospiraceae), as well as certain Actinobacteria, were observed, correlating positively with elevated levels of fecal monosaccharides, such as fructose and glucose. This association suggests increased degradation of complex carbohydrates into simple, absorbable sugars. The accumulation of these sugars has been linked to elevated levels of pro-inflammatory cytokines. In contrast, microbial profiles associated with insulin sensitivity (IS) displayed a higher abundance of Alistipes, Bacteroides, and Faecalibacterium, which correlated negatively with HOMA-IR and fecal sugar levels. Furthermore, they favored the production of fermentative metabolites, such as SCFAs. These findings were experimentally validated by the oral administration of *Alistipes indistinctus* in murine models, which significantly improved insulin sensitivity and reduced intestinal monosaccharide load [55].

Studies in obese pediatric populations have reinforced the association between IR and gut dysbiosis. In this group, a significant reduction in the phylum Firmicutes and a relative increase in Bacteroidetes were observed, resulting in a decreased (F/B) ratio compared to insulin-sensitive children. At the taxonomic level, the highest abundance of Coriobacteriales, Turicibacterales, Pasteurellales, and Turicibacteraceae was associated with the IS group, while Peptococcaceae predominated in subjects with IR. In addition, genera such as Butyricimonas, Alistipes, and Anaerostipes showed significant correlations with key biomarkers, such as ANGPTL4 and adropine, suggesting that gut microbial composition not only reflects metabolic status but actively participates in its regulation through cytokine-mediated mechanisms and metabolite production [80]. These findings position the gut microbiota not only as a biomarker of metabolic status but as an active regulator of insulin sensitivity and a potential target for precision clinical interventions.

Another study reported that individuals with IR exhibited an enrichment in the bacterial genera of Lachnospiraceae, i.e., Dorea and Blautia, as well as Actinobacteria, which were correlated with the metabolism of disaccharides and oligosaccharides [55]. Moreover, gut dysbiosis induced by the insulin receptor antagonist S961 led to a reduction in bacterial diversity, increased intestinal permeability, a higher abundance of pro-inflammatory Proteobacteria (Enterobacteriaceae), decreased abundances of Bacteroidetes, Actinobacteria, and Firmicutes, hyperglycemia, and insulin resistance, which were related to impaired insulin signaling that leads to chronic low-grade inflammation [81].

Furthermore, multiple studies have documented significant alterations in the gut microbiota of individuals with ASD, characterized by a state of intestinal dysbiosis involving imbalances in microbial composition and function. This dysbiosis often manifests as an inverted F/B ratio and consistent variations in the abundance of key bacterial genera. Among the most increased taxa in individuals with ASD are *Clostridium* spp., Alistipes [55,82], Desulfovibrio [82,83,84], Sutterella [82], *Akkermansia muciniphila* [50,75,82], and *Prevotella* spp. [77,84]. In contrast, genera considered beneficial, such as Bacteroides, *Bifidobacterium* spp., *Lactobacillus* spp., *Faecalibacterium prausnitzii*, *Roseburia* spp., and *Subdoligranulum* spp., tend to be decreased [50,83]. These alterations impact essential functions, such as the production of neuroactive metabolites (propionic acid, butyrate, and p-cresol), mucin degradation, the regulation of neurotransmitters, such as GABA and glutamate, and the maintenance of intestinal barrier integrity [85,86].

Notably, recent multi-omics studies have identified an elevated fecal GABA/glutamate ratio as a metabolic signature of mild ASD, linked to an overrepresentation of Escherichia spp. Moreover, behavioral alterations observed in mice colonized with E. coli from ASD donors suggest a functional link between microbial neurotransmitter imbalance and ASD symptomatology, pointing to promising diagnostic and therapeutic avenues targeting microbial GABAergic signaling [87].

The loss of butyrate-producing bacteria, such as *Faecalibacterium prausnitzii* and *Roseburia* spp., can compromise epithelial integrity, increase intestinal permeability, and promote the activation of neuro-inflammatory and immune responses. At the same time, the increase in potentially pro-inflammatory bacteria, such as *Clostridium* spp. or *Desulfovibrio* spp., could contribute to the exacerbation of gastrointestinal and neurobehavioral symptoms characteristic of ASD. Taken together, these findings reinforce the modulatory role of the gut microbiota in the pathophysiology of ASD, operating through the microbiota–gut–brain axis, with relevant clinical implications for the neurological, immunological, and digestive homeostasis of these patients [88,89].

Although no specific studies have examined dysbiosis profiles exclusively in the context of IR and ASD, indirect comparisons have been made using research focused on related metabolic conditions, such as obesity and type 2 diabetes. This correlation is especially relevant given that individuals with ASD have been reported to be at increased risk of developing obesity, more so than diabetes, because of factors such as selective food intake, sedentary lifestyle, and prolonged use of antipsychotics [90].

A comparison of the microbial profiles observed in ASD and in conditions associated with IR reveals common patterns of bacterial alterations, highlighting increases in genera such as Alistipes, Desulfovibrio, and Prevotella, as well as decreases in *Bifidobacterium* spp. and butyrate-producing bacteria (Figure 2) (Table 1). This convergence suggests that manipulating the gut microbiota through dietary interventions and prebiotic or probiotic strategies could represent a promising therapeutic approach to improve not only ASD symptoms but also metabolic alterations associated with IR in this vulnerable population [55,82].

Additionally, another study reported the presence of specific *Clostridium* species, non-spore-forming anaerobes, and microaerophilic bacteria in children with ASD compared to controls, further supporting the potential role of dysbiosis in the onset of autism [91]. Moreover, a large metagenomic study of 1627 children revealed that ASD is associated with alterations not only in bacteria but also in archaea, fungi, viruses, and microbial functions. A multikingdom marker panel achieved high diagnostic accuracy, suggesting the gut microbiome’s potential as a non-invasive tool for ASD detection [92].

**Table 1 ijms-26-06537-t001:** Differential abundance and functional implications of key gut microbial taxa in IR and ASD.

Bacteria	Alteration in ASD	Functional Role or Effect	Alteration in IR	References
*Clostridium* spp.	Increased	Includes species affecting immunity and metabolism; p-cresol producers	Increased	[82,93,94]
*Bacteroides* spp.	Decreased	Beneficial commensal; reduction may impair intestinal barrier function	Decreased	[82,93,95]
Alistipes	Increased	May disrupt cognition via propionic acid production	Increased	[55,82]
Desulfovibrio	Increased	Sulfate-reducing bacteria; may induce mucosal damage and neuroinflammation	Increased	[82,83,84]
Sutterella	Increased	Associated with gastrointestinal symptoms in ASD	-	[82]
*Akkermansia muciniphila*	Increased	Mucin-degrading bacteria; affects mucus barrier integrity	Decreased	[50,75,82]
*Bifidobacterium* spp.	Decreased	Psychobiotic and SCFA producer; modulates GABA and glutamate	Decreased	[76,82]
*Lactobacillus* spp.	Decreased	Psychobiotic; modulates gut–brain axis communication	-	[95]
*Prevotella* spp.	Increased	Fiber-degrading bacteria; increased after microbiota transfer therapy (MTT)	Increased	[77,84]
*Faecalibacterium prausnitzii*	Altered	Butyrate producer; regulates immune function	Decreased	[50,83]
*Roseburia* spp.	Altered	Butyrate producer; supports epithelial tight junction integrity	-	[50]
*Subdoligranulum* spp.	Decreased	Butyrate producer; reduced in ASD	-	[50]
Bacteroidetes	Decreased	Important for polysaccharide digestion; reduction may allow overgrowth of other bacteria	Increase	[80,96,97]
Firmicutes	Decreased	F/B ratio inversion; implications for neurodevelopment and inflammation	Decreased	[80,98]

## 5. Therapeutic and Research Implications

IGF-1 is a neurotrophic molecule essential for CNS development, promoting neuronal proliferation, migration, survival, and synapse formation. It has emerged as a promising therapeutic candidate for both syndromic and non-syndromic forms of ASD due to its critical role in CNS development and function [26,99]. Preclinical models of Rett syndrome have demonstrated that IGF-1 can ameliorate respiratory and behavioral abnormalities [100]. Moreover, a study on the impact of IGF-1 treatment on neurons derived from ASD patients compared to controls revealed heterogeneous responses among ASD patients, depending on the levels of IGF-1 receptor expression [26]. These findings suggest that IGF-1 could represent a targeted therapy to address core neural deficits in ASD.

Building on these findings, a double-blind, placebo-controlled Phase II trial (ClinicalTrials.gov Identifiers: NCT01970345) was initiated to evaluate the safety and feasibility of IGF-1 in children with non-syndromic ASD [101]. Although the trial was terminated in April 2023 because of drug supply issues, the convergence of molecular evidence on IGF1R signaling underscores a shared mechanistic axis in ASD and opens new avenues for targeted metabolic therapies.

In parallel, the convergence of immunometabolic mechanisms suggests microbiota-targeted therapies as a promising therapeutic option for neurodevelopmental disorders such as ASD. For example, *Bacteroides fragilis* BF839 showed efficacy in a controlled clinical trial, improving gastrointestinal symptoms and behavior in children with ASD [102]. Another study, involving microbiota transfer therapy in children diagnosed with ASD, reported beneficial changes in the gut environment, including increased abundance of Bifidobacterium, Prevotella, and Desulfovibrio, leading to improvements in both gastrointestinal and behavioral symptoms of ASD that persisted for up to two years post-treatment [103]. Complementary interventions with probiotics aimed at enhancing SCFA-producing bacteria (Bifidobacterium, Lactobacilllus) have also shown benefits for both insulin resistance via metabolic anti-inflammation and ASD symptomatology [104,105]. Collectively, these findings support therapeutic strategies that restore gut barrier function, thereby reducing central neuroinflammation.

## 6. Future Directions

There is increasing evidence that gut dysbiosis is linked to IR, systemic low-grade chronic inflammation, and neurodevelopmental disorders such as ASD. Future case–control studies should assess the underlying molecular pathways involved in microbiota–host interactions. A promising approach would be the characterization of the specific microbial signatures and metabolites that modulate host immune responses through TLRs and other PRRs. High-throughput metagenomics and metabolomics could facilitate the identification of bacterial taxa and microbial-derived metabolites that either attenuate or exacerbate pro-inflammatory cytokine production in both peripheral and central tissues. These findings could help to classify patients based on their microbial profiles and inflammatory signatures, enabling the development of microbiota-targeted therapeutic interventions.

Another area of research could be the investigation of the developmental and maternal origins of gut–brain axis disruption, particularly during gestation and early life. Given that the maternal microbiome could play a pivotal role in fetal immune and neural development, studies using animal models should explore whether maternal insulin resistance or gut dysbiosis promotes the translocation of bacterial endotoxins, such as LPSs, into fetal brain tissue. This research could clarify whether inflammatory signaling cascades (TLR4-MyD88-NF-κB, RAGE-PI3K/Akt/mTOR) are initiated in utero, potentially predisposing offspring to ASD-like neurodevelopmental phenotypes. These findings could be critical for initiating maternal monitoring and early preventive strategies.

Moreover, strategies of intervention through dietary and microbial modulation should be assessed in animal models with a risk of metabolic or neurodevelopmental disorders. These studies could evaluate the impact of prebiotics, probiotics, postbiotics, or specific dietary interventions on microbial composition, intestinal permeability, and systemic inflammation, as well as their effect on IR and ASD pathogenesis.

Finally, the identification of specific inflammatory biomarkers of ASD could enhance the personalized medicine approach. Biomarkers, such as serum LPS, IL-6, or TNF-α, and microbial metagenomic and metabolite profiling, could be integrated into diagnostic frameworks to detect early-stage dysbiosis to predict disease risk. Furthermore, longitudinal studies are needed to assess whether changes in the gut microbiota, and systemic inflammation are related to the onset of neurological symptoms in a gut–immune–brain axis mode, potentially guiding the development of novel diagnostic and therapeutic strategies.

## 7. Conclusions

This review consolidates emerging evidence supporting a functional convergence between gut microbiota dysbiosis, insulin resistance, and autism spectrum disorder. The changes in the gut microbiota are characterized by an increase in pro-inflammatory bacteria and a decrease in beneficial, anti-inflammatory genera. This suggests that the gut microbiota plays a significant role in both metabolic and neurodevelopmental disorders. Bacterial metabolites, such as lipopolysaccharides, trimethylamine N-oxide, and phenylacetic acid, may act as mediators that trigger immune activation and disrupt insulin signaling.

The identified parallels between autism spectrum disorder and insulin resistance in terms of microbial signatures and inflammatory pathways highlight the importance of investigating the gut–brain–metabolic axis. These insights open new avenues for microbiota-targeted interventions, including probiotics, dietary modulation, and anti-inflammatory strategies, particularly in individuals with autism spectrum disorder who have metabolic comorbidities, such as obesity or insulin resistance.

## Figures and Tables

**Figure 1 ijms-26-06537-f001:**
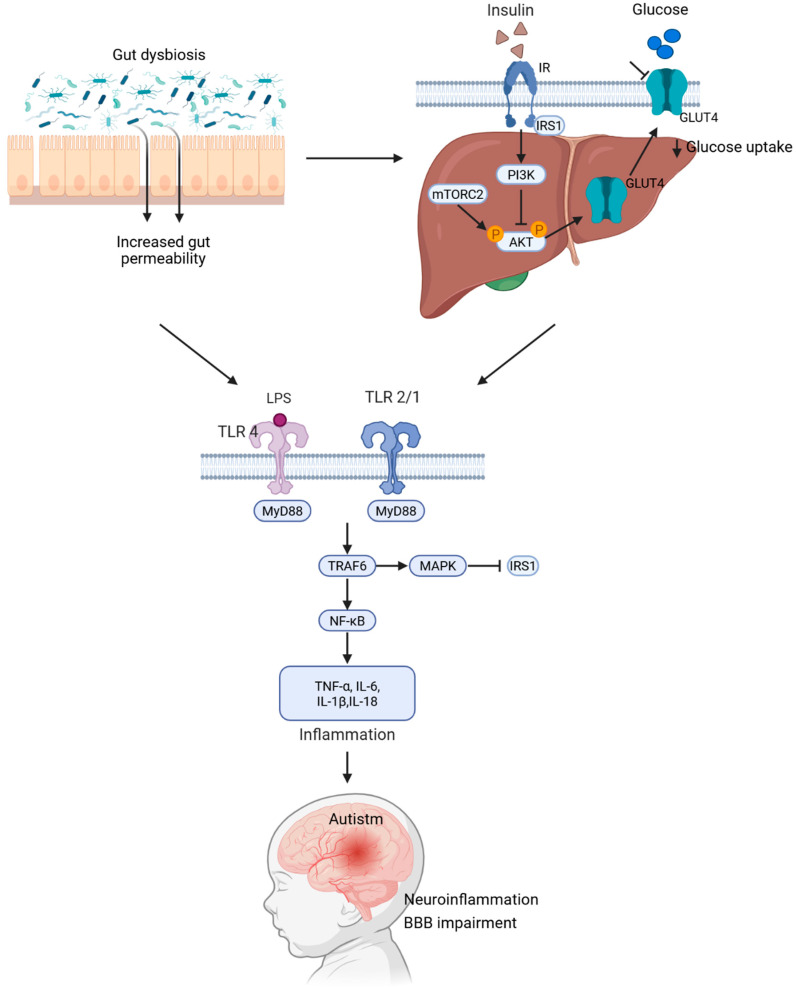
Mechanistic model linking gut dysbiosis, insulin resistance, and autism spectrum disorder. Gut dysbiosis leads to increased intestinal permeability, allowing lipopolysaccharides to enter the bloodstream. This triggers the activation of TLR4 and TLR2/1 through MyD88. Consequently, this initiates the TRAF6–MAPK–NF-κB signaling pathway, resulting in an elevation in pro-inflammatory cytokines, such as TNF-α, IL-6, IL-1β, and IL-18. These cytokines induce inhibitory phosphorylation of IRS1, which dampens the PI3K/AKT-GLUT4 pathway and contributes to systemic insulin resistance. The resulting chronic inflammation and disruption of the blood–brain barrier promote neuroinflammation and neurodevelopmental changes associated with ASD.

**Figure 2 ijms-26-06537-f002:**
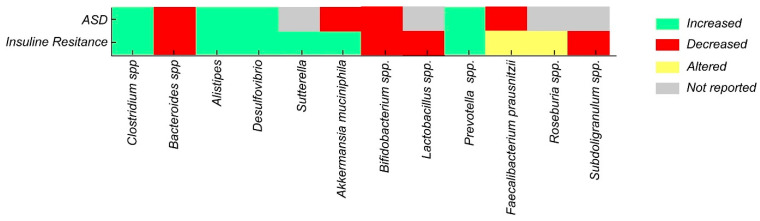
Alterations in gut bacterial taxa in ASD vs. IR.

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
