# Peer review of "Mechanistic Links Between Gut Dysbiosis, Insulin Resistance, and Autism Spectrum Disorder"

_ijms, 2025, doi:10.3390/ijms26136537_

Round 1

Reviewer 1 Report

Comments and Suggestions for Authors
  1. The manuscript presents valuable associations, but lacks a unifying conceptual model that clearly articulates how gut dysbiosis and IR mechanistically converge to influence ASD. A schematic diagram summarizing this triad is strongly recommended to guide the reader.
  2. Multiple sections revisit the same signaling pathways (e.g., TLR/NF-κB, PI3K/Akt/mTOR) without adding new insights. The manuscript would benefit from clearer section organization that distinguishes:
    • The effects of dysbiosis in ASD
    • The role of IR in neurodevelopment
    • Overlapping molecular mechanisms
    • Therapeutic and research implications
  3. Many claims are presented as fact without discussing the strength or limitations of the cited studies. For instance, the clinical link between IR and ASD is largely speculative and based on limited evidence. The authors should critically appraise the available data and avoid overstating conclusions.
  4. Statements about maternal dysbiosis leading to fetal brain colonization or ASD onset are intriguing but speculative. These should be clearly framed as hypotheses and not presented as established mechanisms unless supported by direct evidence.
  5. The manuscript would be significantly improved by:
    • A mechanistic diagram illustrating key pathways (TLR/NF-κB, LPS, SCFA, etc.)
    • A consolidated comparative table of microbiota changes in ASD and IR
    • A figure summarizing potential therapeutic targets or intervention points
  6. Numerous grammatical errors and awkward phrasings are present throughout the manuscript. A professional English language edit is strongly advised to enhance clarity and readability.
  7. The future research section is promising but underdeveloped. The authors should elaborate on:
    • The potential of high-throughput metagenomics and metabolomics
    • Clinical trials targeting the microbiota in ASD-IR comorbidity
    • Use of maternal models to evaluate in utero microbial influences

Besides,

  • The manuscript title may be too broad. Consider revising it to emphasize the molecular link or mechanistic overlap between IR and ASD via gut dysbiosis.
  • Ensure consistent use of terminology (e.g., IR vs. insulin resistance, ASD vs. autism).
  • The abstract could be more concise and focused, summarizing key findings and implications.

Comments on the Quality of English Language
  1. Numerous grammatical errors and awkward phrasings are present throughout the manuscript. A professional English language edit is strongly advised to enhance clarity and readability.

Author Response

Thank you for your comments and suggestions. All the authors are grateful for your time and effort.

  1. The manuscript presents valuable associations, but lacks a unifying conceptual model that clearly articulates how gut dysbiosis and IR mechanistically converge to influence ASD. A schematic diagram summarizing this triad is strongly recommended to guide the reader.

Thank you for your suggestion. We have added a schematic figure that illustrates the relationship between intestinal dysbiosis, insulin resistance, and autism spectrum disorder.

  1. Multiple sections revisit the same signaling pathways (e.g., TLR/NF-κB, PI3K/Akt/mTOR) without adding new insights. The manuscript would benefit from clearer section organization that distinguishes:
  • The effects of dysbiosis in ASD
  • The role of IR in neurodevelopment
  • Overlapping molecular mechanisms
  • Therapeutic and research implications

Thank you for this observation. We have reorganized the manuscript's structure to enhance clarity and minimize redundancy. The revised sections now follow a logical flow: (1) Introduction, (2) the effects of dysbiosis in ASD, (3) The role of IR in neurodevelopment, (4) Gut Dysbiosis Induces Low-Grade Inflammation and Insulin Resistance, (5) Differential Abundance of Gut Microbial in ASD and Insulin Resistance, (6) Therapeutic and research implications and (7) Future Directions.

  1. Many claims are presented as fact without discussing the strength or limitations of the cited studies. For instance, the clinical link between IR and ASD is largely speculative and based on limited evidence. The authors should critically appraise the available data and avoid overstating conclusions.

Thank you for your comment. We have revised the manuscript to include a critical appraisal of the cited evidence, particularly in the section discussing the IR-ASD link.

  1. Statements about maternal dysbiosis leading to fetal brain colonization or ASD onset are intriguing but speculative. These should be clearly framed as hypotheses and not presented as established mechanisms unless supported by direct evidence.

Thank you for your observation. We have revised these statements throughout the manuscript, clearly framing them as hypotheses and indicating that evidence remains preliminary, especially in human studies.

  1. The manuscript would be significantly improved by:
  • A mechanistic diagram illustrating key pathways (TLR/NF-κB, LPS, SCFA, etc.)
  • A consolidated comparative table of microbiota changes in ASD and IR
  • A figure summarizing potential therapeutic targets or intervention points

Thank you for your valuable suggestions. We have added a schematic figure illustrating the interaction between gut dysbiosis, insulin resistance, and autism spectrum disorder, including key molecular pathways. Additionally, we have retained and refined the comparative table summarizing microbial alterations observed in both ASD and IR.

  1. Numerous grammatical errors and awkward phrasings are present throughout the manuscript. A professional English language edit is strongly advised to enhance clarity and readability.

We appreciate your recommendation. The manuscript has been carefully revised to improve clarity, consistency, and overall readability throughout the text.

  1. The future research section is promising but underdeveloped. The authors should elaborate on:
  • The potential of high-throughput metagenomics and metabolomics
  • Clinical trials targeting the microbiota in ASD-IR comorbidity
  • Use of maternal models to evaluate in utero microbial influences

Thank you for the suggestion. We have expanded the future research section and Therapeutic and research implications section to address these points.

Besides,

  • The manuscript title may be too broad. Consider revising it to emphasize the molecular linkor mechanistic overlap between IR and ASD via gut dysbiosis.

Thank you for the recommendation. We have revised the title to “Mechanistic Links Between Gut Dysbiosis, Insulin Resistance, and Autism Spectrum Disorder.”

  • Ensure consistent use of terminology (e.g., IR vs. insulin resistance, ASD vs. autism).

Thank you, terminology has been standardized throughout the manuscript to ensure consistency.

  • The abstract could be more concise and focused, summarizing key findings and implications.

Thank you. We have revised the abstract to make it more concise.

Reviewer 2 Report

Comments and Suggestions for Authors

Linking Gut Dysbiosis and Insulin Resistance to Autism Spectrum Disorder-A Molecular Perspective

A brief summary

Autism Spectrum Disorder (ASD) is increasingly linked to systemic metabolic issues, with evidence suggesting that gut microbiota dysbiosis and insulin resistance (IR) may interact through shared inflammatory and signaling pathways such as TLR4/NF-κB and PI3K/Akt/mTOR. Both ASD and IR exhibit similar microbial imbalances, which may disrupt neuroimmune function and point to a gut–brain–metabolic axis that could guide future microbiota-based therapies.

General concept comments

Article:

This review offers valuable insights by highlighting the interconnected roles of gut microbiota, insulin resistance, and neuroinflammation in ASD, presenting a comprehensive view of the gut–brain–metabolic axis. By integrating molecular and microbial evidence, the review uncovers shared mechanisms underlying ASD and metabolic dysfunction, such as disruptions in TLR4/NF-κB and PI3K/Akt/mTOR signaling pathways. These findings not only enhance our understanding of ASD etiology but also open new avenues for targeted interventions. The identification of overlapping microbial signatures in ASD and insulin resistance underscores the potential of microbiota-based therapies, which could improve both metabolic health and neurodevelopmental outcomes. Ultimately, this review lays a foundation for more personalized and biologically informed treatment strategies for individuals with ASD.

Review:

To enhance the depth and relevance of the review, it would be valuable to incorporate recent multi-omics studies and systematic reviews exploring the interplay between the gut microbiome, metabolism, and neurodevelopment in ASD. For example:

  • A 2024 systematic review analyzing 44 studies found consistent alterations in gut microbiota composition—particularly involving Firmicutes and Pseudomonadota—in children with ASD, with probiotic interventions showing improvements in both gastrointestinal and behavioral symptoms (El Mazouri S, Aanniz T, Bouyahya A, et al. Gut Microbiota in Autism Spectrum Disorder: A Systematic Review. Progress In Microbes & Molecular Biology. 2024;7(1). DOI: 10.36877/pmmb.a0000442).
  • Zhao, L., Cai, G., Fu, J., Zhang, W., Ye, Y., Ling, Z., & Ye, S. (2024). Gut microbial ‘TNFα–sphingolipids–steroid hormones’ axis in children with autism spectrum disorder: an insight from meta-omics analysis. Journal of Translational Medicine, 22(1), 1165. proposed a novel “TNFα–sphingolipids–steroid hormones” axis, linking dysbiosis, lipid metabolism, and inflammation using integrative meta-omics profiling in ASD.
  • In another recent large-scale metagenomic study of over 1,600 children, researchers identified multi-kingdom microbial signatures—spanning bacteria, archaea, and functional genes—that enabled accurate ASD classification with an AUC of 0.91 (Su Q, Wong OWH, Lu W, Wan Y, Zhang L, Xu W, Li MKT, Liu C, Cheung CP, Ching JYL, Cheong PK, Leung TF, Chan S, Leung P, Chan FKL, Ng SC. Multikingdom and functional gut microbiota markers for autism spectrum disorder. Nature Microbiology. July 9, 2024. DOI: 10.1038/s41564-024-01739-1).
  • Additionally, a 2025 metabolomic analysis (Wang D, Jiang Y, Jiang J, Pan Y, Yang Y, Fang X, Liang L, et al. Gut microbial GABA imbalance emerges as a metabolic signature in mild autism spectrum disorder linked to overrepresented Escherichia. Cell Reports Medicine. 2025 Jan 21;6(1):101919. DOI: 10.1016/j.xcrm.2024.101919) reports that an elevated gut GABA/glutamate ratio is a key metabolic marker of mild ASD and highlights the association between Escherichia overgrowth and disruptions in GABA metabolism, highlighting new metabolic pathways involved in ASD pathophysiology.
  • Finally, the study Aspects of Insulin Resistance in Children with Autism (Journal of IMAB. (2025). Aspects of insulin resistance in children with autism. Journal of IMAB, 31(2), 6111-6115. https://www.journal-imab-bg.org/issues-2025/issue2/vol31issue2p6111-6115.html) provides direct clinical evidence supporting the high prevalence of metabolic dysfunction in ASD, reinforcing the link between insulin signaling and neurodevelopmental outcomes.

Including such studies offers a broader framework for understanding the gut–brain–metabolic axis and supports the development of targeted microbiome-based interventions.

Specific comments:

  • Instead of Table 1, I would suggest using a figure, perhaps a histogram
  • I would suggest using the STRING web server (https://string-db.org/) to analyze pathways of groups of genes and their functional enrichment.

Author Response

We appreciate your valuable feedback and the time you invested in evaluating our work. Your suggestions have been very helpful in improving the quality of the manuscript.

To enhance the depth and relevance of the review, it would be valuable to incorporate recent multi-omics studies and systematic reviews exploring the interplay between the gut microbiome, metabolism, and neurodevelopment in ASD. For example:

  • A 2024 systematic review analyzing 44 studies found consistent alterations in gut microbiota composition—particularly involving Firmicutes and Pseudomonadota—in children with ASD, with probiotic interventions showing improvements in both gastrointestinal and behavioral symptoms (El Mazouri S, Aanniz T, Bouyahya A, et al. Gut Microbiota in Autism Spectrum Disorder: A Systematic Review. Progress In Microbes & Molecular Biology. 2024;7(1). DOI: 10.36877/pmmb.a0000442).
  • Zhao, L., Cai, G., Fu, J., Zhang, W., Ye, Y., Ling, Z., & Ye, S. (2024). Gut microbial ‘TNFα–sphingolipids–steroid hormones’ axis in children with autism spectrum disorder: an insight from meta-omics analysis. Journal of Translational Medicine, 22(1), 1165. proposed a novel “TNFα–sphingolipids–steroid hormones” axis, linking dysbiosis, lipid metabolism, and inflammation using integrative meta-omics profiling in ASD.
  • In another recent large-scale metagenomic study of over 1,600 children, researchers identified multi-kingdom microbial signatures—spanning bacteria, archaea, and functional genes—that enabled accurate ASD classification with an AUC of 0.91 (Su Q, Wong OWH, Lu W, Wan Y, Zhang L, Xu W, Li MKT, Liu C, Cheung CP, Ching JYL, Cheong PK, Leung TF, Chan S, Leung P, Chan FKL, Ng SC. Multikingdom and functional gut microbiota markers for autism spectrum disorder. Nature Microbiology. July 9, 2024. DOI: 10.1038/s41564-024-01739-1).
  • Additionally, a 2025 metabolomic analysis (Wang D, Jiang Y, Jiang J, Pan Y, Yang Y, Fang X, Liang L, et al. Gut microbial GABA imbalance emerges as a metabolic signature in mild autism spectrum disorder linked to overrepresented Escherichia. Cell Reports Medicine. 2025 Jan 21;6(1):101919. DOI: 10.1016/j.xcrm.2024.101919) reports that an elevated gut GABA/glutamate ratio is a key metabolic marker of mild ASD and highlights the association between Escherichia overgrowth and disruptions in GABA metabolism, highlighting new metabolic pathways involved in ASD pathophysiology.
  • Finally, the study Aspects of Insulin Resistance in Children with Autism (Journal of IMAB. (2025). Aspects of insulin resistance in children with autism. Journal of IMAB, 31(2), 6111-6115. https://www.journal-imab-bg.org/issues-2025/issue2/vol31issue2p6111-6115.html) provides direct clinical evidence supporting the high prevalence of metabolic dysfunction in ASD, reinforcing the link between insulin signaling and neurodevelopmental outcomes.

Including such studies offers a broader framework for understanding the gut–brain–metabolic axis and supports the development of targeted microbiome-based interventions.

Thank you for the excellent references. We have integrated these recent studies into the relevant sections of the manuscript to strengthen the discussion on the interplay between the gut microbiome, metabolism, and neurodevelopment in ASD.

Specific comments:

  • Instead of Table 1, I would suggest using a figure, perhaps a histogram

Thank you for the suggestion. We have decided to retain Table 1, as it allows for a clear comparison of microbial alterations in ASD and IR. However, to enhance visual interpretation, we have also added a diagram (Fig 2) that illustrates the distribution of bacterial taxa differences between both conditions.

  • I would suggest using the STRING web server (https://string-db.org/) to analyze pathways of groups of genes and their functional enrichment.

Thank you for the recommendation. At this stage, we did not perform STRING analysis, as our focus was on reviewing current evidence rather than conducting new network-based analyses. We consider it a valuable approach for future work.

Reviewer 3 Report

Comments and Suggestions for Authors

In this narrative review, the authors tried to link ASD with insulin resistance.

  • Some questions remain unsolved. While there are numerous evidences indicating a correlation between autism and gut dysbiosis, as well as between insulin resistance (IR) and gut dysbiosis, a correlation between ASD and IR seems too speculative, or at least indirect, not further corroborate by literature.
  • The table 1 provided, indeed, only refers to ASD and gut alterations, no indication between ASD and IR is provided.
  • Could a structured review or meta-analysis (PRISMA) provide better description ?
  • The title refers to a molecular point of view, however in the text less is given on this aspect.
  • ASD increased incidence is due not only to better diagnosis, current evidences support a novel role of environment.
Comments on the Quality of English Language

English requires editing, as example lines 78 - 82 are not very clear.

Author Response

Thank you for your comments and support. We are grateful for your careful review and helpful insights.

In this narrative review, the authors tried to link ASD with insulin resistance.

  • Some questions remain unsolved. While there are numerous evidences indicating a correlation between autism and gut dysbiosis, as well as between insulin resistance (IR) and gut dysbiosis, a correlation between ASD and IR seems too speculative, or at least indirect, not further corroborate by literature.

Thank you for your feedback. We've added a section titled "The Role of Insulin Resistance in Neurodevelopment," which explores recent evidence supporting a possible connection between insulin resistance and ASD and discusses their shared mechanisms.

  • The table 1 provided, indeed, only refers to ASD and gut alterations, no indication between ASD and IR is provided.

Thank you for your feedback. Table 1 has been updated to include microbial alterations observed in both TEA and IR, highlighting relevant overlaps. Moreover, we have also added a diagram (Fig 2) that illustrates the distribution of bacterial taxa differences between both conditions.

  • Could a structured review or meta-analysis (PRISMA) provide better description ?

We appreciate your suggestion. Although a systematic review in PRISMA format was not conducted, we employed a targeted search strategy and critically selected relevant studies to support the manuscript's approach.

  • The title refers to a molecular point of view, however in the text less is given on this aspect.

Thank you for your feedback. The relevant section has been restructured and the molecular pathways involved have been described more clearly, aligning the content with the focus of the title. Moreover, we change the title to “Mechanistic Links Between Gut Dysbiosis, Insulin Resistance, and Autism Spectrum Disorder”.

  • ASD increased incidence is due not only to better diagnosis, current evidences support a novel role of environment.

Thanks for the suggestion. We've incorporated the role of environmental factors into the manuscript.

Round 2

Reviewer 1 Report

Comments and Suggestions for Authors

The authors have thoroughly addressed my previous comments, and the revised manuscript is now suitable for publication.

Reviewer 3 Report

Comments and Suggestions for Authors

Authors made significant efforts to improve their manuscript.